# Molecular Mechanisms Regulating the Columnar Tree Architecture in Apple

Kazuma Okada [1] and Chikako Honda [2],*

1 Institute of Fruit Tree and Tea Science, National Agriculture and Food Research Organization, 2-1 Fujimoto, Ibaraki 305-8605, Japan; okak@affrc.go.jp
2 Graduate School of Agricultural and Life Sciences, The University of Tokyo, Midoricho, Tokyo 188-0002, Japan
* Correspondence: hondac@g.ecc.u-tokyo.ac.jp

**Abstract:** The columnar apple cultivar 'McIntosh Wijcik' was discovered as a spontaneous mutant from the top of a 'McIntosh' tree in the early 1960s. 'McIntosh Wijcik' exhibits the columnar growth phenotype: compact and sturdy growth, short internodes, and very few lateral shoots. Classical genetic analysis revealed that the columnar growth phenotype of 'McIntosh Wijcik' is controlled by a single dominant gene, *Co*. This review focuses on the advances made toward understanding the molecular mechanisms of columnar growth in the last decade. Molecular studies have shown that an 8.2 kb insertion in the intergenic region of the *Co* locus is responsible for the columnar growth phenotype of 'McIntosh Wijcik', implying that the insertion affects the expression patterns of adjacent genes. Among the candidate genes in the *Co* region, the expression pattern of *MdDOX-Co*, putatively encoding 2-oxoglutarate-dependent dioxygenase (DOX), was found to vary between columnar and non-columnar apples. Recent studies have found three functions of *MdDOX-Co*: facilitating bioactive gibberellin deficiency, increasing strigolactone levels, and positively regulating abscisic acid levels. Consequently, changes in these plant hormone levels caused by the ectopic expression of *MdDOX-Co* in the aerial organs of 'McIntosh Wijcik' can lead to dwarf trees with fewer lateral branches. These findings will contribute to the breeding and cultivation of new columnar apple cultivars with improved fruit quality.

**Keywords:** abscisic acid; *Co* locus; dioxygenase; gibberellin; insertion; *Malus*; mutant; mutation; plant hormone; strigolactone

## 1. Introduction

Tree architecture influences various aspects of fruit production and orchard management [1], such as planting density, fruit quality and yield, pruning and training, fruit thinning and harvesting, and pesticide spraying. Tree architecture is regulated by four processes: primary growth, branching patterns, flowering location, and meristem and shoot mortality [1]. Recently, molecular mechanisms regulating tree architecture have been extensively studied [2–4].

Apple (*Malus × domestica* Borkh.) trees have been classified under four tree architectural types (types I to IV) according to both the overall tree growth pattern and their fruiting type: type I (columnar type), type II (spur type), type III (standard type), and type IV (tip-bearing type; this differs from the genetic weeping type [5]) (Figure 1A–E) [1]. Columnar apples (type I) display multiple distinct traits: compact and sturdy growth, short internodes, very few lateral shoots, many spurs, and large and thick leaves with high chlorophyll content (Figure 1A,B,F) [6–8]. Consequently, the tree grows naturally as a column.

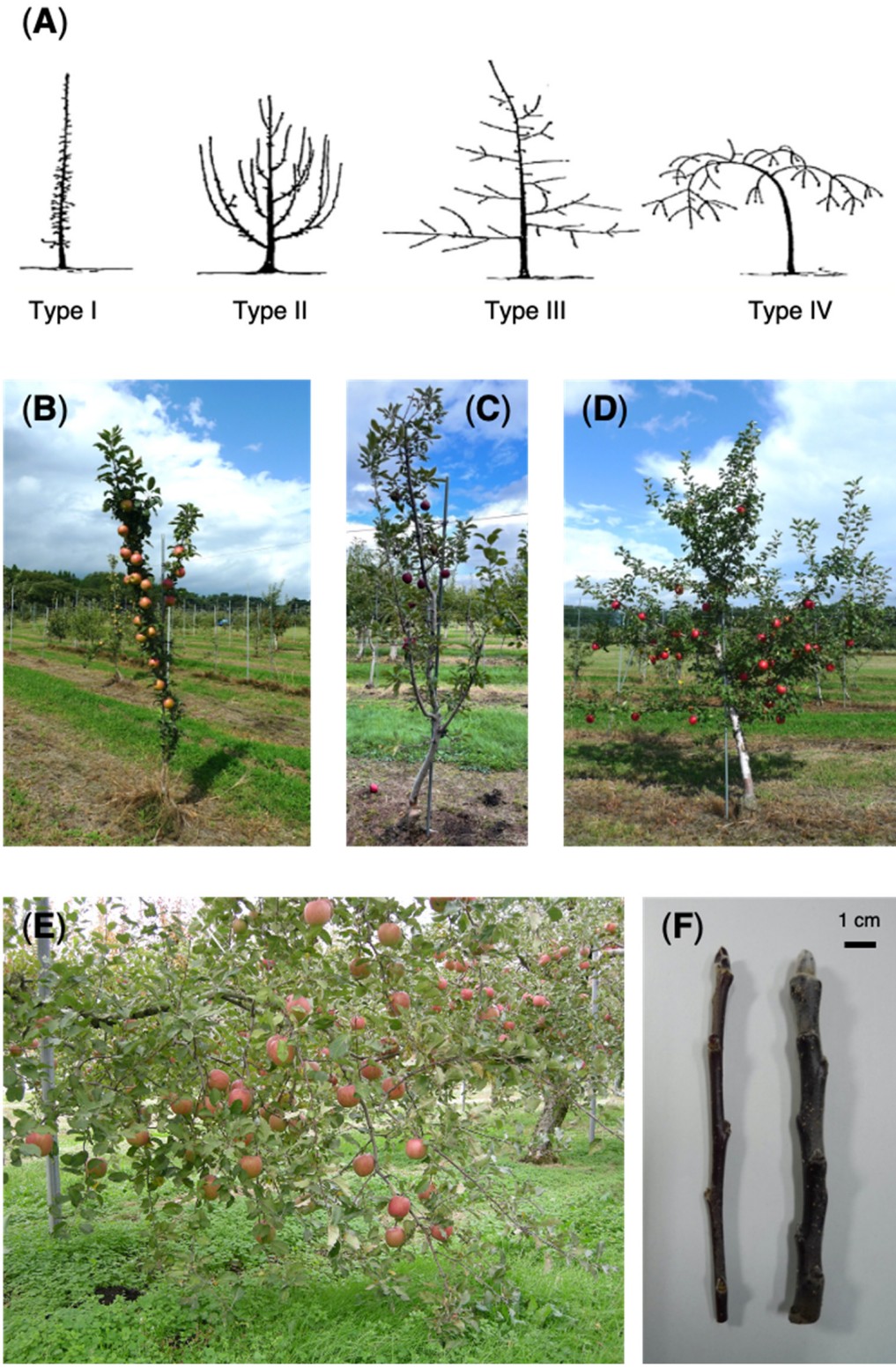

**Figure 1.** Apple tree architectures and one-year-old branches. (**A**) Tree architectures of four types (type I to IV) of apple; reproduced with permission from Costes et al. [1]. (**B**) Type I (columnar type, seedling); reproduced with permission from Okada [9]. (**C**) Type II (spur type, 'Wellspur Delicious'). (**D**) Type III (standard type, seedling); reproduced with permission from Okada [9]. (**E**) Type IV (tip-bearing type, 'Fuji'). (**F**) One-year-old branches of 'McIntosh' (left) and 'McIntosh Wijcik' (right); reproduced with permission from Okada [10].

The columnar architecture of apple trees was first discovered in 'McIntosh Wijcik'. 'McIntosh Wijcik' was identified as a spontaneous mutant arising from the top of a 50-year-old 'McIntosh' tree in the early 1960s. It exhibits very slow growth, a negligible number of side shoots, compact and upright growth habits, and biennial bearing [11,12]. Genetic analysis revealed that the columnar growth phenotype of 'McIntosh Wijcik' is controlled by a single dominant gene, *Co* [6]. In addition, a possible role of certain modifier genes was suspected because the percentage of compact seedlings obtained from the test crosses was consistently lower than expected [6,13]. Since dominant genes controlling tree architecture in apple are valuable, 'McIntosh Wijcik' has been used not only as an important genetic resource to develop compact cultivars for high-density orchards but also as a model system to elucidate the mechanism of apple tree architecture determination [13].

The columnar growth phenotype permits high-density planting, minimal pruning, and mechanical harvesting [2,8]. Columnar apples can also function as space-saving pollinizers in orchards of a single cultivar because apples exhibit self-incompatibility [2,14]. The agronomic characteristics of columnar apple cultivars have been studied from the perspective of fruit production. Owing to the suppressed growth of side branches, columnar apples can be planted at 1 m intervals [11]. Moreover, Inomata et al. [15] compared the fruit productivity of 'Maypole' columnar apple trees which were trained in a central leader system at 0.66 m intervals with that of trees trained in a Y-trellis system at 1.14 m intervals. They concluded that a Y-trellis system was more advantageous than a central leader system because apple trees trained in a Y-trellis system maintained a lower fruiting position and displayed higher dry matter and fruit productivity. However, columnar apple trees generally suffer from susceptibility to biennial bearings [11]. Blazek and Krelinova [16] reported biennial bearing in five new columnar apple cultivars, most likely because of the proximately between the *Co* locus and biennial bearing quantitative trait loci (QTLs) in the shared linkage group [17]. Iwanami et al. [18] proposed a new labor-saving method for processing columnar apple trees using a thinning strategy for biennial bearings.

Recently, significant advances have been made toward the understanding of the molecular mechanisms of columnar growth in apple trees. In this review, we describe the genetic, molecular, physiological, and biochemical features of columnar apples elucidated over the last decade.

## 2. Fine Mapping of the *Co* Locus

Previously, the *Co* locus was mapped on linkage group 10 of 'McIntosh Wijcik' [19]. However, the causative mutation and *Co* candidate genes remain unknown. Since the release of a high-quality draft genome sequence of the apple cultivar Golden Delicious by Velasco et al. [20], several research groups have performed fine mapping of the *Co* locus using novel DNA markers developed from apple genome sequence information (Figure 2A). Firstly, Bai et al. [21] delimited the *Co* locus between markers C1753-3520 and C7629-22009, and the physical size was estimated to be 193 kb in the 'Golden Delicious' genome. Thereafter, Moriya et al. [22] restricted the *Co* locus between markers Mdo.chr10.11 and Mdo.chr10.15, with a size of 196 kb in the apple genome. Third, Baldi et al. [23] also narrowed down the *Co* locus between markers Co04R10 and Co04R13, and the determined size was 393 kb in the apple genome. Fourth, Morimoto and Banno [24] defined the *Co* locus between markers LG10-Co-N and C7629_12936 as 530 kb apart in the apple genome. Finally, Okada et al. [25] further delimited the *Co* locus between the markers Mdo.chr10.11-2 and Mdo.chr10.13-2, with a distance of 101 kb between them in the apple genome. From these results, the position of the *Co* locus may be deduced between the markers Co04R10 and C7629-22009, and the physical size of this region (genomic coordinates: 18.51–19.10 Mb) on chromosome 10 of the 'Golden Delicious' genome is estimated to be ca. 590 kb.

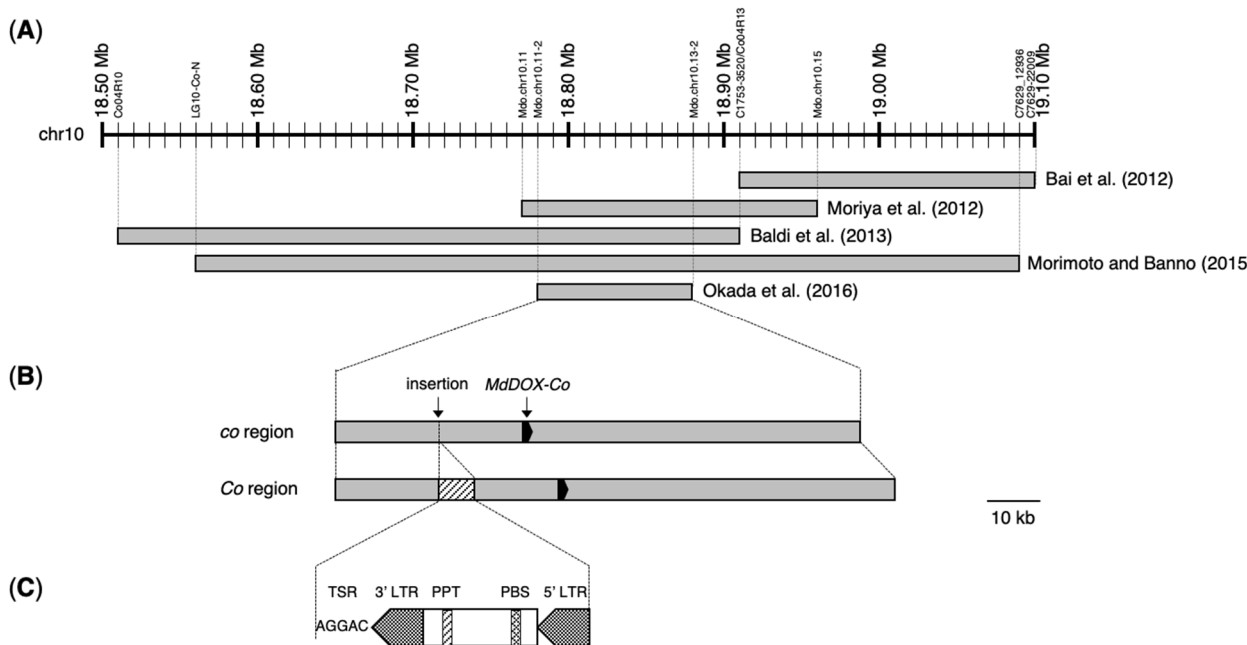

**Figure 2.** Identifying an insertional mutation by positional cloning. (**A**) Fine mapping of the *Co* locus (gray bars) on chromosome 10. (**B**) Difference of genomic structures between the *co* and *Co* regions. *MdDOX-Co* is located approximately 16 kb downstream of the insertion. (**C**) Structure of the 8.2 kb insertion sequence. LTR, long terminal repeat; PBS, primer binding site; PPT, polypurine tract; TSR, target site repeat. Reproduced with data from [21–25].

## 3. Identification of Mutation in 'McIntosh Wijcik'

To identify the mutation responsible for the tree architecture of 'McIntosh Wijcik', three research groups used a positional cloning approach. Wolters et al. [26,27] sequenced BAC clones encompassing the *Co* region of 'McIntosh Wijcik' (*Co/co*) and the *co* region of 'McIntosh' (*co/co*). Similarly, Otto et al. [28] cloned and sequenced the *Co* region from the columnar cultivar Procats 28 (*Co/co*) and the homologous genomic regions of 'McIntosh' and 'McIntosh Wijcik'. Okada et al. [25] also sequenced BAC clones containing the *Co* region of the columnar cultivar Telamon (*Co/co*) and the *co* region of 'McIntosh'. Three research groups have unanimously reported that an 8.2 kb insertion in the *Co* region is the only genomic difference between the *Co* and *co* regions (Figure 2B). Therefore, the columnar growth phenotype in 'McIntosh Wijcik' is attributed to this insertional mutation.

The insertion consists of two long terminal repeats (LTRs) of 1951 bp each, a primer binding site (PBS), and a polypurine tract (PPT) required for transposition, but lacks the typical ORFs encoding the group-specific antigen (Gag), reverse transcriptase, integrase, or RNase H [25,28], exhibiting the characteristics of a non-autonomous LTR retroposon (Figure 2C). The 100% sequence identity between the two LTRs indicates that the insertion sequence had translocated recently, supporting that the mutation in 'McIntosh Wijcik' occurred approximately 60 years ago [28].

## 4. Exploration of *Co* Candidate Genes

Because the insertional mutation in 'McIntosh Wijcik' was found in an intergenic region instead of the coding region, it was hypothesized that the expression patterns of adjacent genes may be affected rather than the disruption of the coding region of any gene [26]. To identify the *Co* candidate gene, Wolters et al. [26] analyzed the expression levels of six genes (*MdCo27–MdCo32*) predicted in the 50 kb region encompassing the insertion sequence using quantitative reverse transcription PCR. They revealed that only *MdCo31* was upregulated (14-fold) in the axillary buds of 'McIntosh Wijcik' compared to

'McIntosh'. However, the expression of *MdCo31* was not observed in the leaves of both 'McIntosh Wijcik' and 'McIntosh'.

Otto et al. [28] and Petersen et al. [29] mapped the RNA-seq data obtained from shoot apical meristems (non-columnar cultivar A14-190-93 and columnar cultivar Procats 28), leaves ('McIntosh' and 'McIntosh Wijcik'), and primary roots (non-columnar, heterozygous columnar, and homozygous columnar apples) to the *Co* region. Furthermore, they analyzed the expression levels of five genes located in the vicinity of the insertion, as well as that of the insertion sequence, using RNA-seq and quantitative real-time PCR. The expression levels of *MDP0000927098* (*ATL5K-like*), *MDP0000163720* (*ACC1-like*), and the insertion sequence were elevated in the shoot apical meristems of the columnar apples. Moreover, *downy mildew resistance 6-like* (*dmr6-like*) was strongly upregulated in the shoot apical meristems and newly developing leaves of columnar apples. However, it was downregulated in the primary roots of heterozygous columnar apples and remained suppressed in the young but fully developed leaves of both cultivars. *At1g08530-like* and *MDP0000934866* (*At1g06150-like*) showed similar expression levels in all tissues of both the columnar and non-columnar apples.

Okada et al. [25] mapped the RNA-seq data of shoot apices of 'McIntosh Wijcik' and 'McIntosh' to the *Co* region and analyzed differentially expressed genes. Five contigs (12053, 41231, 38029, 44905, and 91071-genes) were upregulated in 'McIntosh Wijcik', consistent with the fact that the *Co* gene is dominant. In contrast, a single contig (18023-gene) was downregulated in 'McIntosh Wijcik'. Homology searches using BLAST showed that, among the six contigs, the 91071-gene was the most likely candidate for the *Co* gene. Furthermore, Okada et al. [25] showed that the three genes (*MDP0000927098*, *MDP0000163720*, and *MDP0000934866*) identified previously by Otto et al. [28] and Petersen et al. [29] did not show differential expression between 'McIntosh Wijcik' and 'McIntosh'.

Overall, a common *Co* candidate gene was identified by three research groups and given different names: *MdCo31* [26], *dmr6-like* [29], and 91071-gene [25]. Therefore, the *MdCo31/dmr6-like*/91071-gene is the strongest candidate for the *Co* gene. In this review, the designation '*MdDOX-Co*' [30] is used instead of the aforementioned names.

## 5. Expression Analysis of *MdDOX-Co*

In non-columnar apples, *MdDOX-Co* was primarily expressed in the roots, whereas no or negligible expression was noted in the shoot apices, axillary buds, and leaves. In contrast, columnar apples expressed *MdDOX-Co* in the roots, shoot apices, axillary buds, and leaves [26,28,29,31–34]. In situ hybridization showed that *MdDOX-Co* is expressed in the growing root tips and lateral root primordium of both non-columnar and columnar apples and in the shoot meristem and leaf primordium of 'McIntosh Wijcik' [31]. A negligible difference was observed in the expression levels and patterns of *MdDOX-Co* between the roots of non-columnar and columnar apples [31]. In addition, the normal tree architecture of non-columnar scions grafted onto columnar rootstocks indicated that the columnar growth phenotype was not transmissible from rootstock to scion [31]. These results suggest that ectopic expression of *MdDOX-Co* in aerial organs (shoot apices, axillary buds, and leaves) is responsible for columnar growth, whereas its expression in the roots is not associated with columnar growth.

## 6. Phenotypes of Transgenic Plants Overexpressing *MdDOX-Co*

To investigate the effects of *MdDOX-Co* on phenotype, transgenic plants overexpressing *MdDOX-Co* were generated. *Arabidopsis* plants overexpressing *MdDOX-Co* displayed compact plants with dark green leaves and short floral internodes [26,35]. Moreover, tobacco plants overexpressing *MdDOX-Co* showed decreased plant height and internode length, thick and wrinkled leaves with high chlorophyll content, and delayed flowering [25,30,32–34]. Similarly, apples overexpressing *MdDOX-Co* exhibited short internodes and suppressed upward growth [25]. These results confirm that *MdDOX-Co* expression

leads to short plant height and internode length in various plants, and the short internodes of columnar apples result from the upregulated expression of *MdDOX-Co.*

## 7. Characterization of *MdDOX-Co*

*MdDOX-Co* encodes a putative 2-oxoglutarate-dependent dioxygenase (DOX) consisting of 339 amino acids that belongs to the DOXC class of the DOX superfamily. The members of this family participate in various oxygenation/hydroxylation reactions in plants [25,26,36]. Transient expression in *Arabidopsis* cells demonstrated that the MdDOX-Co-GFP fusion protein was specifically localized in the cytoplasm [34]. Phylogenetic analysis of DOXs classified MdDOX-Co into the DOXC41 clade along with hyoscyamine 6β-hydroxylase, which catalyzes the hydroxylation of hyoscyamine, and *Hordeum vulgare* iron-deficiency-specific clones 2 and 3, involved in the biosynthesis of mugineic acid [25,36–38]. Because these phytochemicals are apparently unrelated to tree architecture, the mechanism by which *MdDOX-Co* influences the columnar growth phenotype remains unclear.

## 8. Functions of *MdDOX-Co* and the Mechanisms of the Columnar Growth Phenotype

Recently, three functions have been proposed for *MdDOX-Co*:

(1) *MdDOX-Co* causes bioactive gibberellin (GA) deficiency.

Okada et al. [30] closely examined tobacco plants overexpressing *MdDOX-Co*, which showed a typical dwarf phenotype (short plant height and internode length, wide leaf shape, and dark green wrinkled leaves) and were similar to the GA- or brassinosteroid (BR)-deficient/signaling mutants. GA is a plant hormone involved in stem elongation, seed germination, and flowering [39], whereas BR is related to cell elongation, cell division, and cell differentiation [40]. The dwarf phenotypes of transgenic tobacco plants were restored to traits similar to the wild-type plants after the application of gibberellin $A_3$ ($GA_3$), but not after the application of brassinolide [30,34]. Similarly, 'McIntosh Wijcik' showed dwarf traits (short main shoot and internode length), which were partially reversed by $GA_3$ application. Interestingly, $GA_3$ treatment of young apple trees also increases the number of lateral branches and reduces the number of flower buds [30]. Furthermore, reduced endogenous concentrations of bioactive GAs ($GA_1$ and/or $GA_4$) were observed in transgenic tobacco plants and columnar apples compared with those in wild-type tobacco plants and non-columnar apples, respectively [30,34]. These results suggest that ectopic expression of *MdDOX-Co* in the transgenic tobacco plants and the aerial organs of columnar apples causes a dwarf phenotype, which is mediated by bioactive GA deficiency, and that GA plays an important role in controlling apple tree architecture by promoting vegetative growth (shoot and internode elongation and lateral branch formation) and inhibiting reproductive growth (flower bud formation).

Recently, Watanabe et al. [35] demonstrated that recombinant MdDOX-Co metabolizes $GA_{12}$, $GA_9$, and $GA_4$ to $GA_{111}$, $GA_{70}$, and $GA_{58}$, respectively (Figure 3). In addition, exogenous application of $GA_{12}$ to *Arabidopsis* plants overexpressing *MdDOX-Co* produced $GA_{111}$, but not $GA_{70}$, $GA_{58}$, $GA_9$, or $GA_4$. They also confirmed a lower efficiency of conversion of $GA_{111}$ to $GA_{70}$ by recombinant *Arabidopsis* GA 20-oxidases than that of the conversion of $GA_{12}$ to $GA_9$. These data indicate that the conversion of $GA_{12}$ to $GA_{111}$ by MdDOX-Co blocks the pathway of production of biologically active GAs ($GA_4$ and $GA_{58}$).

Okada et al. [30] and Watanabe et al. [35] proposed that the bioactive GA deficiency caused by the ectopic expression of *MdDOX-Co* in the aerial organs of 'McIntosh Wijcik' can lead to early growth cessation and short internodes in both the main and side shoots. This possibly resulted in the formation of a dwarf tree with very few lateral branches and numerous spurs. Consequently, the synergistic action of these pleiotropic traits manifests as a columnar tree form.

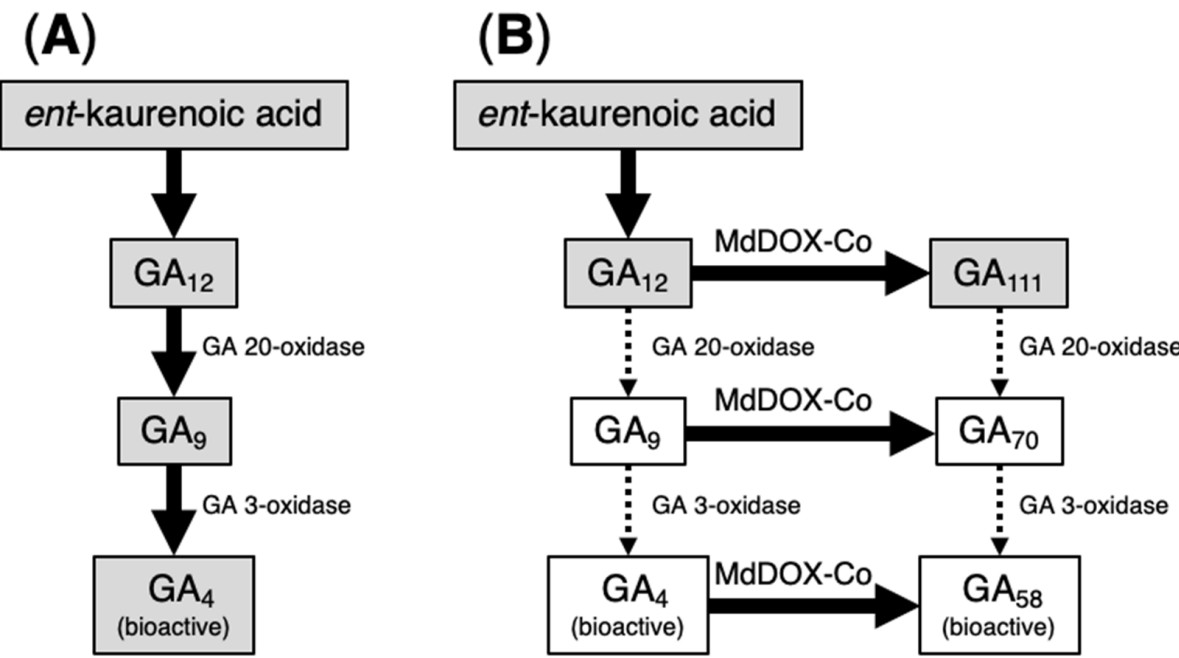

**Figure 3.** Effect of MdDOX-Co on the biosynthesis of bioactive gibberellin (GA). (**A**) Simplified $GA_4$ biosynthesis pathway in plants. (**B**) MdDOX-Co metabolizes $GA_{12}$, $GA_9$, and $GA_4$ to $GA_{111}$, $GA_{70}$, and $GA_{58}$, respectively; conversion of $GA_{12}$ to $GA_{111}$ by MdDOX-Co prevents the biosynthesis of bioactive GAs. Reproduced with permission from Okada [9].

(2) *MdDOX-Co* increases strigolactone (SL) content.

SLs are plant hormones implicated in the inhibition of bud outgrowth and shoot branching [41]. To study the relationship between SLs and the columnar growth pheno-type, Sun et al. [32] characterized the expression profiles of SL biosynthesis- and signal transduction-related genes in columnar and non-columnar apples. Expression levels of the major genes involved in SL biosynthesis, including the *MORE AXILLARY GROWTH* genes (*MdMAX3-1* and *MdMAX4-4*) and a *DWARF* gene (*MdD27-1*), were higher in both buds and shoots of columnar apples than in the corresponding tissues of non-columnar apples. Expression level of the *DWARF* gene *MdD53-4*, which represses SL signal transduction, was lower in columnar apples. In addition, tobacco plants overexpressing *MdDOX-Co* showed a higher expression level of *NbMAX3* and lower expression level of *NbD53* than wild-type plants. Because the SL content in columnar apples was higher than that in non-columnar apples, it may be inferred that *MdDOX-Co* increased the SL content, weakened the inhibition of SL signal transduction, and inhibited lateral branching in columnar apples.

(3) *MdDOX-Co* positively regulates abscisic acid (ABA) biosynthesis and enhances salt tolerance.

ABA is a stress-response hormone that inhibits shoot and lateral bud elongation [33]. The ABA content in shoots of 'McIntosh Wijcik' was significantly higher than that in shoots of 'McIntosh', and the ABA content of 35S:*MdDOX-Co* transgenic apple calli was also higher than that of wild-type (non-transgenic non-columnar apple) calli [33]. Furthermore, the expression levels of the major ABA biosynthesis genes, *MdNCED6* and *MdNCED9* (*MdNCED6/9*), were significantly higher in the 'McIntosh Wijcik' and 35S:*MdDOX-Co* apple calli than in the 'McIntosh' and wild-type calli, respectively. Sun et al. [33] further indicated that MdDOX-Co forms a protein complex (MdDOX-Co–MdDREB2) with transcription factor MdDREB2. MdDREB2 directly binds to *cis*-elements in the *MdNCED6/9* promoters, thereby functioning as a transcriptional activator. The expression levels of *MdNCED6/9* and the ABA content were higher in transgenic apple calli co-overexpressing *MdDOX-Co* and *MdDREB2* than in transgenic plants overexpressing *MdDOX-Co* or *MdDREB2* inde-pendently [33]. These results suggest that the MdDOX-Co–MdDREB2 complex promotes

ABA biosynthesis by upregulating the expression of *MdNCED6/9*. Thus, *MdDOX-Co* plays a positive role in ABA biosynthesis, and the higher ABA content in columnar apples may lead to two effects: (1) ABA directly reduces the elongation of lateral buds, resulting in a few lateral branches, and (2) ABA suppresses the effect of GA, with consequent inhibition of branch and internode growth, ultimately leading to a dwarf phenotype [33].

Sun et al. [42] also found that the expression level of *MdDOX-Co* increased remarkably in 'McIntosh Wijcik' under salt stress. In addition, shoot cultures of 'McIntosh Wijcik' exhibited higher salt tolerance than those of 'McIntosh'. Transgenic tobacco and apple calli overexpressing *MdDOX-Co* also displayed enhanced salt tolerance, higher superoxide dismutase activity, and lower malondialdehyde levels than wild-type plants under salt stress [42]. Thus, *MdDOX-Co* was concluded to confer salt tolerance.

## 9. Modifier Genes and Other Genes Involved in the Columnar Growth Phenotype

Dougherty et al. [43] identified two recessive loci (*c2* and *c3*) that can suppress the columnar growth phenotype, and *c2* appeared to have a more prominent effect in younger (2-year-old) trees than in older (8-year-old) trees. The *c2* locus is located on chromosome 10 and the *c3* locus is located on chromosome 9, and these two loci were suggested to repress the columnar growth phenotype through additive gene interactions. Trees with a repressed columnar growth phenotype (phenotype, standard growth; genotype, *Coco*) showed a drastic reduction in the expression level of *MdDOX-Co*.

Transcriptome analyses also revealed that many differentially expressed genes were associated with the columnar growth phenotype. Zhang et al. [44] identified 5237 differentially expressed genes between newly developing shoots of columnar and non-columnar apples using RNA-seq. Among the 5237 unigenes, the gene ontology functional annotation and KEGG pathway database identified 287 unigenes related to plant architecture formation. Among the 287 unigenes, 106 were GRAS transcription factors, suggesting that GRAS transcription factors play an important role in regulating the architecture of apple trees.

Similarly, Krost et al. [45,46] compared the transcriptomes of shoot apical meristems of columnar apple 'Procats 28' and standard apple 'A14-190-93' using RNA-seq. Genes that were categorized into cell wall modification, transport, and protein modification were upregulated in the columnar apple, whereas genes that were grouped into light reactions, mitochondrial electron transport, lipid metabolism, cell wall, DNA synthesis, RNA processing, and protein synthesis were downregulated in the columnar apple. Furthermore, 16 plant hormone-associated genes were differentially regulated: indole-3-acetic acid (6 genes), cytokinin (3 genes), ABA (3 genes), BR (2 genes), GA (1 gene), and jasmonic acid (1 gene). Their regulation probably leads to an increase in the endogenous bioactive indole-3-acetic acid, cytokinin, BR, GA, and jasmonic acid in the columnar apple [46].

Otto et al. [28] also mapped the RNA-seq data of leaves from 'McIntosh' and 'McIntosh Wijcik' to the 'Golden Delicious' genome. A total of 5751 of 9961 unigenes were differentially expressed between 'McIntosh' and 'McIntosh Wijcik'. Genes that are involved in secondary metabolism (especially lignin and terpene biosynthesis), metabolism and/or signaling of plant hormones (auxin, jasmonate, and ethylene), glutathione-*S*-transferases, and proteins that are involved in defense or stress reactions (such as pathogen recognition receptors and heat shock proteins) were upregulated in leaves of 'McIntosh Wijcik'. In contrast, genes that are associated with photosynthesis, protein biosynthesis, and nucleotide metabolism and enzymes managing the redox state (such as thioredoxin, dismutase/catalase, and peroxidase) were downregulated in leaves of 'McIntosh Wijcik'.

## 10. Marker-Assisted Selection (MAS) Systems for Selecting Columnar Apples

'McIntosh Wijcik' serves as an important genetic resource for breeding columnar apple cultivars and has been used in several breeding programs at the East Malling Research Station that were initiated in the early 1970s [47]. Consequently, four columnar cultivars were released for amateur gardeners: an ornamental apple, 'Maypole' ('McIntosh Wijcik' × 'Baskatong'), and three dessert apples, 'Telamon' ('McIntosh Wijcik' × 'Golden Delicious'), 'Trajan' ('Golden

Delicious' × 'McIntosh Wijcik'), and 'Tuscan' ('McIntosh Wijcik' × 'Greensleeves') [12]. However, outstanding commercial columnar cultivars with good fruit quality have not yet been developed, and ongoing breeding programs are aimed at producing such cultivars.

It is difficult to differentiate between columnar and non-columnar apples phenotypically until the seedlings are 2–3 years old [48]. Therefore, MAS systems, which can be applied to several-week-old seedlings, facilitate the selection of columnar apples. Conventional SSR markers linked to the *Co* gene were unable to distinguish the *Co* allele from the original wild-type *co* allele of 'McIntosh' [22,25]. Hence, DNA markers specific to the *Co* allele have been developed based on the insertion polymorphism. Wolters et al. [26] and Otto et al. [28] performed PCR with primer pairs spanning the left or right borders of the 8.2 kb insertion and obtained a fragment of the expected size, specifically from the genomic DNA of columnar cultivars, which was absent in non-columnar cultivars. However, these DNA markers could not discriminate between heterozygous columnar apples (*Co/co*) and homozygous columnar apples (*Co/Co*) when a single PCR was performed. Okada et al. [25] and Cmejlova et al. [48] conducted PCRs using three primers designed from the insertion polymorphism and simultaneously identified homozygous columnar apples (*Co/Co*), heterozygous columnar apples (*Co/co*), and non-columnar apples (*co/co*) through a single PCR. These *Co* allele-specific PCR products are useful DNA markers in the MAS of columnar apples for segregating progenies even when 'McIntosh' or its offspring are used as a parent because it can distinguish the *Co* allele from the original wild-type *co* allele [25,26].

## 11. Conclusions and Perspectives

Extensive research in the last decade has successfully identified the causative mutation (insertion of an 8.2 kb LTR retroposon) and gene (*MdDOX-Co*) for the columnar growth phenotype. Interestingly, MdDOX-Co is involved in GA metabolism [30,34,35], but it is categorized into a different phylogenetic clade (DOXC41) from those containing key enzymes associated with GA biosynthesis and degradation: GA 3-oxidase (DOXC3), GA 20-oxidase (DOXC7), $C_{19}$-GA 2-oxidase (DOXC12), $C_{20}$-GA 2-oxidase (DOXC13), and GA 7-oxidase (DOXC22) [25,35,36]. This suggests that MdDOX-Co is a unique enzyme involved in GA metabolism. Furthermore, *MdDOX-Co* is associated with an increase in SL and ABA biosynthesis [32,33]. Therefore, how *MdDOX-Co* affects the columnar growth phenotype remains to be elucidated, and more detailed analyses are needed. It is important to investigate whether MdDOX-Co is an enzyme with multiple functions or whether the multiple functions result from crosstalk among plant hormones. In addition, how modifier genes (*c2* and *c3*) reduce the expression level of *MdDOX-Co* needs to be elucidated in future studies.

Columnar apples have labor-saving properties for apple growers and fit well with sustainable high-density planting, resulting in higher yields per hectare. However, an important problem is that columnar apple cultivars with superior fruit quality have not yet been developed [48]. The columnar growth phenotype can also easily be combined with other desirable characteristics (e.g., disease resistance) by crossing because the columnar growth phenotype is controlled by the dominant *Co* gene [47]. Recent *Co* allele-specific DNA markers enable the efficient selection of seedlings with a columnar growth phenotype and will facilitate the breeding of new columnar apple cultivars with good fruit quality and other desirable characteristics.

Another important problem is the development of new cultivation systems suitable for the columnar apples, as the tree architecture of columnar apples is completely different from that of standard apples (Figure 1). New cultivation systems include mechanization, which leads to labor savings. Mechanical harvesters that can take advantage of the flat arrangement of branches on columnar apples are particularly promising. As a promising cultivation method to control tree size, exogenous $GA_3$ application is an efficient way to optimize tree height in columnar apples [30]. $GA_3$ application in the juvenile phase will promote tree growth in columnar apples, whereas the cessation of $GA_3$ application in the adult phase will maintain a desirable tree form that hardly needs pruning and training.

Thus, the creation of a series of high quality new columnar apple cultivars, as well as innovative cultivation systems involving mechanization, may dramatically revolutionize future apple industry as in the case of the invention of high-yielding semi-dwarf varieties of wheat and rice that led to the "Green Revolution" [49].

**Author Contributions:** Writing—original draft preparation, K.O.; writing—review and editing, C.H. All authors have read and agreed to the published version of the manuscript.

**Funding:** Our work was supported by JSPS KAKENHI grant numbers 24780033, 21H02126 and by the Ministry of Agriculture, Forestry and Fisheries of Japan grant number HOR-2002.

**Institutional Review Board Statement:** Not applicable.

**Informed Consent Statement:** Not applicable.

**Acknowledgments:** We thank Masato Wada and Takashi Haji for providing photos of Figure 1C,E, respectively. We are grateful to Masato Wada and Masatoshi Nakajima for carefully proofreading the manuscript.

**Conflicts of Interest:** The authors declare no conflict of interest.

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
