# Peer review of "Molecular Mechanisms Regulating the Columnar Tree Architecture in Apple"

_forests, doi:10.3390/f13071084_

Round 1
Reviewer 1 Report
As apple research experts, the authors should have done a lot of research on columnar apple. The two Chinese researchers (Bai Tuanhui and Zhang Yuyang) studied on columnar apple I consulted both were cited in current review. The author should have spent valuable energy in summarizing the review on the molecular mechanism of columnar apple in the past decade, and the summary is very clear.
It is suggested that the contents of the text "11. columnar apple production" be incorporated into the Introduction as a brief introduction for "columnar apple". In the Abstract, the sentence “In addition, several important findings about columnar apple production have been discussed at the end of the review” should be removed, and mention the content of 9 (other genes involved in columnar growth) and 10 (MAS systems for selecting columnar apples) from main text in the Abstract.
Author Response
It is suggested that the contents of the text "11. columnar apple production" be incorporated into the Introduction as a brief introduction for "columnar apple".
Thank you for your comment. According to it, the content of section 11 was moved to the introduction section.
In the Abstract, the sentence “In addition, several important findings about columnar apple production have been discussed at the end of the review” should be removed, and mention the content of 9 (other genes involved in columnar growth) and 10 (MAS systems for selecting columnar apples) from main text in the Abstract.
The abstract was revised to correspond to the above revision.
Reviewer 2 Report
This manuscript was a very straightforward, well-organized review about all of the known biology associated with the columnar apple tree architecture. Overall, I think this is a very comprehensive and useful paper. The references cited and researched described was appropriate. The figures were also useful in exemplifying the content. I found it helpful to have a clear visual of the location of the different mappings for the gene in figure 2 (which was blurry in the review copy), and Figure 3 was a good visual of where DOX-co fits in. I suggest providing the apple variety for the images in Figure 1C-E if available. Also, I ‘m not sure I would consider 1E to be a type IV, or rather it is unclear. From what I understand, the weeping described by type IV is more of a tip bearer and the fruit weight promotes some of the weeping. The image there is of a weeping apple but It might be weeping from a specific gene mutation that isn’t commonly found in apples that are considered type IV. I believe Honeycrisp is type IV (and maybe Fuji ?) and those certainly don’t look like 1E.
There were a few sentences with minor grammar, wording, or clarity issues, including but not limited to those listed below. I suggest having someone outside of the authors give the paper one final read for clarity to catch other mistakes if present.
1. Abstract sentence: “Consequently, changes in these plant hormones that caused by ectopic expression”. Change to “that were caused by”
2. Introduction sentence: “Tree architecture consists of four processes..” – I would rephrase to say something like “Tree architecture is regulated by four processes…” “Or “is a result of four processes”
3. Introduction last sentence: What is meant by space saving pollinizer? Do mean for orchards? That phrasing was unclear. Also high-density apple orchards do not need space saving pollinizers. Those trees are trained to be narrow.
Author Response
I suggest providing the apple variety for the images in Figure 1C-E if available.
Thank you for your comment. The variety for the image in Figure 1C is ‘Wellspur Delicious’. The apples for the image in Figure 1B, 1D and 1E are seedlings.
Also, I ‘m not sure I would consider 1E to be a type IV, or rather it is unclear. From what I understand, the weeping described by type IV is more of a tip bearer and the fruit weight promotes some of the weeping. The image there is of a weeping apple but It might be weeping from a specific gene mutation that isn’t commonly found in apples that are considered type IV. I believe Honeycrisp is type IV (and maybe Fuji ?) and those certainly don’t look like 1E.
Thank you for your valuable comment. We misinterpreted type IV (tip-bearing type) as weeping. Type IV can be assumed to result from the individual shoot propensity to bend under its own weight and the fruit load (Costes et al. 2006). Therefore, we replaced the image in Figure 1E.
There were a few sentences with minor grammar, wording, or clarity issues, including but not limited to those listed below. I suggest having someone outside of the authors give the paper one final read for clarity to catch other mistakes if present.
Thank you for your comment. The revised manuscript received English editing services again. The parts of the text that were revised as a result of the English editing are indicated in blue.
- Abstract sentence: “Consequently, changes in these plant hormones that caused by ectopic expression”. Change to “that were caused by”
Thank you for your comment. We have revised the manuscript as you indicated.
- Introduction sentence: “Tree architecture consists of four processes.” – I would rephrase to say something like “Tree architecture is regulated by four processes…” “Or “is a result of four processes”
Thank you for your comment. We have revised the manuscript as you indicated.
- Introduction last sentence: What is meant by space saving pollinizer? Do mean for orchards? That phrasing was unclear. Also high-density apple orchards do not need space saving pollinizers. Those trees are trained to be narrow.
Thank you for your comment. We have added the purpose for space-saving pollinizer.